METHODS

# Bayesian clustering with uncertain data

**Kath Nicholls**[1,2], **Paul D. W. Kirk**[1,2,3], **Chris Wallace**[1,2]*

**1** Cambridge Institute of Therapeutic Immunology and Infectious Disease, University of Cambridge, Cambridge, United Kingdom, **2** MRC Biostatistics Unit, University of Cambridge, Cambridge, United Kingdom, **3** Cancer Research UK Cambridge Centre, Ovarian Cancer Programme, University of Cambridge, Cambridge, United Kingdom

* cew54@cam.ac.uk

**Data Availability Statement:** The datasets were derived from sources in the public domain, as follows: • Ferreira from https://www.ebi.ac.uk/arrayexpress/experiments/E-MTAB-1724 • Lyons from https://www.ebi.ac.uk/arrayexpress/experiments/E-MTAB-145 • Chaussabel from

## Abstract

Clustering is widely used in bioinformatics and many other fields, with applications from exploratory analysis to prediction. Many types of data have associated uncertainty or measurement error, but this is rarely used to inform the clustering. We present Dirichlet Process Mixtures with Uncertainty (DPMUnc), an extension of a Bayesian nonparametric clustering algorithm which makes use of the uncertainty associated with data points. We show that DPMUnc out-performs existing methods on simulated data. We cluster immune-mediated diseases (IMD) using GWAS summary statistics, which have uncertainty linked with the sample size of the study. DPMUnc separates autoimmune from autoinflammatory diseases and isolates other subgroups such as adult-onset arthritis. We additionally consider how DPMUnc can be used to cluster gene expression datasets that have been summarised using gene signatures. We first introduce a novel procedure for generating a summary of a gene signature on a dataset different to the one where it was discovered, which incorporates a measure of the variability in expression across signature genes within each individual. We summarise three public gene expression datasets containing patients with a range of IMD, using three relevant gene signatures. We find association between disease and the clusters returned by DPMUnc, with clustering structure replicated across the datasets. The significance of this work is two-fold. Firstly, we demonstrate that when data has associated uncertainty, this uncertainty should be used to inform clustering and we present a method which does this, DPMUnc. Secondly, we present a procedure for using gene signatures in datasets other than where they were originally defined. We show the value of this procedure by summarising gene expression data from patients with immune-mediated diseases using relevant gene signatures, and clustering these patients using DPMUnc.

## Author summary

Identifying groups of items that are similar to each other, a process called clustering, has a range of applications. For example, if patients split into two distinct groups this suggests that a disease may have subtypes which should be treated differently. Real data often has measurement error associated with it, but this error is frequently discarded by clustering methods. We propose a clustering method which makes use of the measurement error

https://www.ebi.ac.uk/arrayexpress/experiments/E-GEOD-11907 We make the implementation of DPMUnc, and the code to generate the analyses in this manuscript, available in github repositories: • DPMUnc software https://github.com/chr1swallace/DPMUnc • DPMUnc simulation study https://github.com/chr1swallace/DPMUnc_simulations • GWAS dataset analysis https://github.com/chr1swallace/DPMUnc_GWAS • Gene expression dataset analysis https://github.com/chr1swallace/clusteringPublicGeneExpr.

**Funding:** This work is supported by the Wellcome Trust (WT220788 to CW, WT220024 to KN) and the Medical Research Council (MC UU 00002/4 to CW, MC UU 00002/13 to PDWK).

**Competing interests:** This work is supported by the Wellcome Trust (WT220788 to CW, WT220024 to KN) and the Medical Research Council (MC UU 00002/4 to CW, MC UU 00002/13 PDWK). The views expressed are those of the author(s) and not necessarily those of the NHS, the NIHR or the Department of Health and Social Care. For the purpose of open access, the authors have applied a CC BY public copyright licence to any Author Accepted Manuscript version arising from this submission. The funders had no role in study design, data collection and analysis, decision to publish, or preparation of the manuscript.

and use it to cluster diseases linked to the immune system. Gene expression datasets measure the activity level of all $\sim 20,000$ genes in the human genome. We propose a procedure for summarising gene expression data using gene signatures, lists of genes produced by highly focused studies. For example, a study might list the genes which increase activity after exposure to a particular virus. The genes in the gene signature may not be as tightly correlated in a new dataset, and so our procedure measures the strength of the gene signature in the new dataset, effectively defining measurement error for the summary. We summarise gene expression datasets related to the immune system using relevant gene signatures and find that our method groups patients with the same disease.

## Introduction

Grouping items by similarity has a variety of applications in bioinformatics. For example, clustering samples according to any molecular measures may help to identify subgroups of disease [1, 2]. In gene expression data, clustering the genes may be useful for inferring gene function using guilt by association and inferring regulatory relationships [3]. Clustering can also be used to cluster diseases themselves. For example, immune-mediated diseases (IMD) have been shown to have a shared genetic basis such that IMD can be clustered according to those shared patterns [4, 5].

Commonly used methods, such as k-means [6, 7] and mclust [8], assume that the observations being clustered are observed without error, or that any errors are identically distributed. In reality we often have not just data points themselves but also a measure of uncertainty about each observation which has the potential to improve clustering accuracy if exploited. Examples include data generated by multiple uncertain experts [9], network clustering where edges have an associated score indicating uncertainty, or the cell type annotation problem in single-cell multiomic data, where different omics layers may assign cells to different clusters. In this work, we focus on quantitative data with associated uncertainty, which may arise for example with random measurement error or the use of summary measures such as the average of three blood pressure readings. To be clear, clustering methods already capture one source of uncertainty as the cluster variance, which describes variability in the location of different objects in the same cluster. Here, when we discuss uncertainty, we will focus on uncertainty attached to each observation, which may vary between observations and/or features. In this case, cluster variance would describe the variability of location of the latent observations which can be thought to represent the "true" locations of the objects whose location we have observed with some measured uncertainty.

A variety of methods have been proposed to deal with uncertainty in observations when clustering, including calculating distance between observations accounting for uncertainty [10], clustering representative objects sampled from the distribution associated with each uncertain observation [11, 12], and adapting the likelihood by integrating over the distribution associated with each uncertain observation [13].

Within clustering approaches generally, the Bayesian methods offer some advantages, in their ability to convey uncertainty about the overall clustering, and particularly to simultaneously infer the number of clusters, $K$. Here, we propose to adapt a standard model for a Bayesian Dirichlet process mixture model to the setting where the data points have associated uncertainty, "Dirichlet Process Mixtures with Uncertainty (DPMUnc)". Such adaptation has not been explored previously, to the best of our knowledge.

We show that accounting for uncertainty can alter the clustering solution returned and demonstrate its performance on a range of simulated datasets and real applications. First, we consider clustering IMD using GWAS summary statistics, a problem where the uncertainty associated with each observation generally varies systematically with study sample size. Second, we cluster patients with IMD using gene expression data which can help identify subtypes of disease or predict response to treatment [2]. However, clustering on all $\sim$ 20,000 genes can be computationally expensive, and may group patients by disease-irrelevant structure, such as sex or age. Investigations of gene expression in relation to disease or biological processes sometimes produce a gene signature, a list of genes with a shared pattern of gene expression which is relevant to the disease or process. We propose a summary measure of a gene signature, which captures average signature expression and its variance, and show that DPMUnc allows patients to be clustered according to one or more signatures.

## Results

### DPMUnc model

Our approach follows previous work [14], extended to include both the observations $x_i$, $i = 1 \ldots n$ and their respective uncertainty estimates $\sigma_i^2$. We model these $x_i$ as noisy observations of some latent variables $z_i$: $x_i \sim \mathcal{N}(z_i, \sigma_i^2)$. The latent variables themselves are modelled as coming from a Dirichlet Process Gaussian Mixture Model (DPGMM). We follow the approach of [15], which takes the infinite limit of a Gaussian Mixture Model and can be shown to be equivalent to using Dirichlet Processes directly [15, 16].

In a finite Gaussian Mixture Model the points are generated from $K$ clusters, where points within each cluster are i.i.d. Gaussian variables with some mean $\mu_k$ and variance $\rho_k^2$:

$$z_i | c_i = k \sim \mathcal{N}(\mu_k, \rho_k^2)$$

where the indicator variables $c_i$ show which cluster a variable belongs to. We give these indicator variables a standard categorical distribution with the weights for each category $\pi_k$ having a Dirichlet prior:

$$p(c_i = k) \quad = \pi_k; \pi_1, \ldots, \pi_K \sim \text{Dirichlet}(\alpha/K, \ldots, \alpha/K)$$

where $\sum_{k=1}^{K} \pi_k = 1$.

Finally, we define priors

$$\alpha \quad \sim \text{Gamma}(a_0, b_0)$$

$$\mu_k, \rho_k^2 \quad \sim \text{Normal-Inverse Gamma}(\mu_0, \kappa_0, \alpha_0, \beta_0)$$

where $a_0 = 3$, $b_0 = 4$, $\mu_0$ is the empirical mean of the observed data and $\alpha_0, \beta_0, \kappa_0$ are hyperparameters that should be adjusted according to the spread of the data and the expected cluster structure. Some further discussion of how these values may be chosen is given in the S1 Text.

The joint density factorises as follows:

$$p(c, x, z, \pi, \mu, \rho, \alpha) = p(\phi)p(c|\pi)p(\pi|\alpha)p(\alpha)\prod_i p(x_i|\sigma_i^2, z_i)p(z_i|\phi_{c_i}) \tag{1}$$

as represented in Fig 1. We now let $K$ tend to infinity, yielding a Gaussian Mixture Model with no constraint on the number of clusters which is equivalent to a Dirichlet Process Gaussian Mixture Model [15, 16].

We use a Gibbs sampling algorithm to sample from the posterior $p(c|x, \sigma^2)$ according to the conditional probability distributions as set out in S1 Text and as described in S1 Fig.

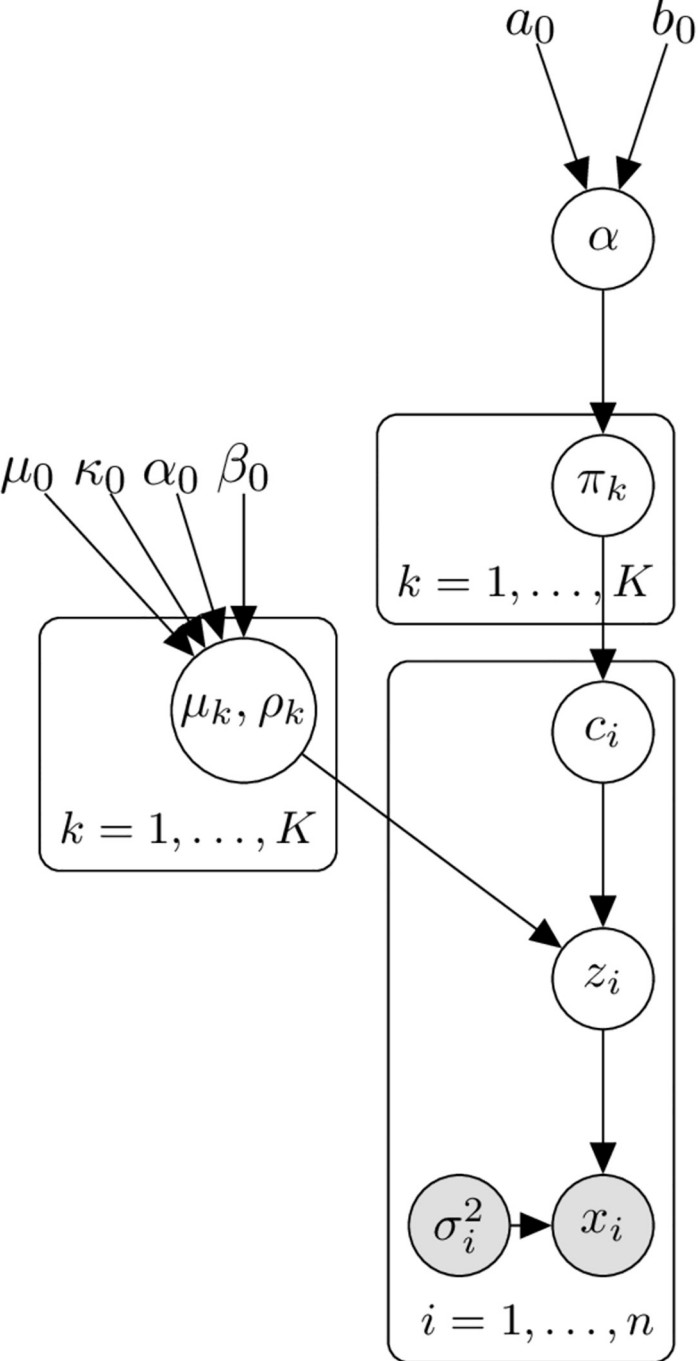

**Fig 1. Plate diagram of Gaussian Mixture Model.** When $K \to \infty$ this is the Dirichlet Process Gaussian Mixture Model that is the basis of our model. In the plate diagram, each rectangle (plate) corresponds to a group of variables, and the text at the bottom of each plate shows how many copies of that plate are required for the full model. Circles indicate random variables, which are shaded if the random variable is observed. The variables without circles are hyperparameters for the model.

This model is stated for univariate data but can easily be extended for multivariate data, using the assumption that dimensions are independent and have the same distribution, so that the same hyperparameters can be used for every dimension. For multivariate data with p dimensions, the variables $x_i$, $z_i$, $\mu_k$ and $\rho_k^2$ would all be p-dimensional vectors and the

corresponding distributions set out in the S1 Text may be used to sample each dimension independently.

To more directly compare the effect of adjusting for the uncertainty of the points, we also run DPMUnc with adjusted versions of the datasets where the observations themselves were untouched, but the uncertainty estimates were shrunk to 0. Thus the rest of the DPMUnc works as normal, but the latent variables are essentially fixed to be equal to the observed data points throughout and so the only difference compared to DPMUnc is that no adjustment for uncertainty takes place. We call this version of the method DPMZeroUnc.

## Simulated study

We first used simulated data to explore the effects of differing levels of uncertainty associated with each observation and noise around each cluster location on DPMUnc. Detailed discussion of two illustrative examples is presented in the S1 Text. Briefly, we found that including estimates of uncertainty tended to produce solutions with smaller cluster variances, and with cluster means corresponding to a weighted mean of observations, in which observations with lower uncertainty were given greater weight, rather than unweighted means when uncertainty was ignored.

We then compared the performance of DPMUnc to a variety of clustering approaches which either do not account for uncertainty in observations:

- kmeans [17], one of the dominant non-parametric approaches that requires the number of clusters, $K$ to be specified. We estimate $K$ using the maximum gap statistic [18]

- mclust [8], based on a mixture of finite Gaussian models. $K$ is chosen as the value which minimises the BIC of a range of models fitted

- DPMZeroUnc, which is DMPUnc with all sample uncertainties set to 0

  or do account for uncertainty:

- hierarchical clustering using the Bhattacharyya distance [19]. We choose $K$ as for kmeans

- fuzcluster [13] which implements a similar model to DPMUnc in a maximum likelihood framework, albeit assuming the number of clusters $K$ is known. We estimate $K$ as for kmeans

- representative clustering of uncertain data [12] which provides a framework to extend a clustering method to uncertain data by sampling multiple examples of possible latent data, applying a clustering method to each and selecting a representative solution. We use representative clustering to extend kmeans and mclust for uncertain data

DPMUnc outperformed its comparators by a small margin when observation noise and cluster variance were small, with the margin increasing with increasing cluster variance or observation noise (Fig 2). kmeans was the best performing of the comparitor methods, and extension with representative clustering did not improve its performance, whilst representative clustering did have a positive impact on the accuracy of mclust. The methods differed in their ability to infer $K$. When the true $K = 3$, DPMUnc and DPMZeroUnc tended to select $K = 3$ or $K = 4$, whilst mclust tended to merge clusters and underestimate $K$, often selecting $K = 1$ as cluster noise increased (S2 Fig), with kmeans and fuzcluster lying between these extremes. Use of representative clustering improved estimation of $K$ for mclust in particular, and restricting $K$ to its true value improved accuracy of mclust and kmeans often to beyond that seen for DPMUnc. This suggests that the DP part of our model is a considerable contributor to its performance.

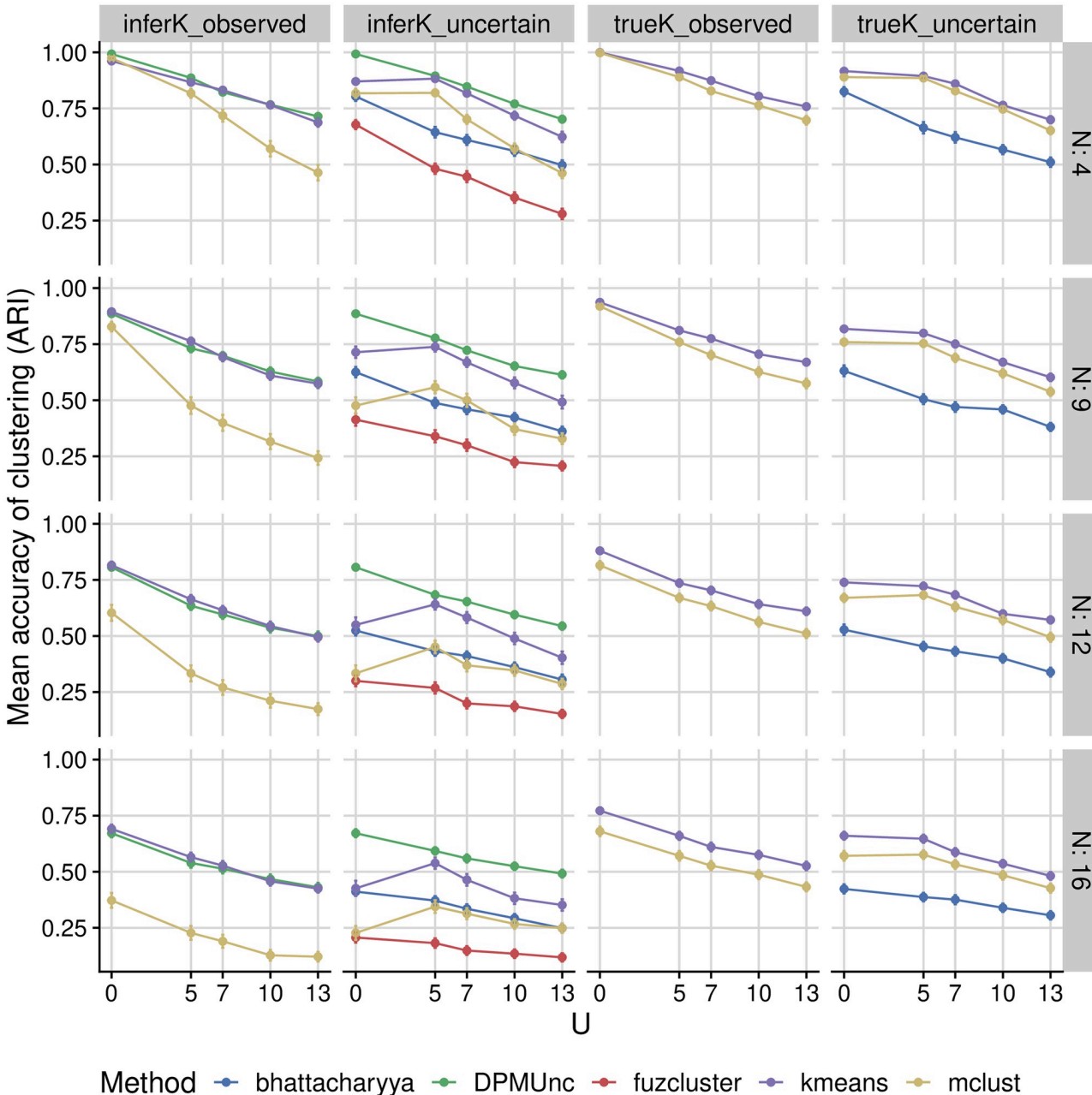

**Fig 2. Accuracy on simulated datasets.** Simulation methods are split according to whether they assumed perfectly observed data or allow for uncertainty. mclust and kmeans do not by default allow for uncertain data, and appear in the second column when used as the clustering engine for representative clustering. The left two columns include results where K is inferred. For methods which allow a known value for *K* to be supplied, we show these for comparison in the righthand two columns. The first row of plots has the lowest noise *N* around the cluster mean, and the bottom row has the highest noise. Increasing uncertainty, corresponding to greater difference between latent data and the observed data is shown on the x-axis. The accuracy of the clustering is given by the Adjusted Rand Index (ARI) between the true clustering and the inferred clustering, averaged over 100 simulations. Higher values of ARI are better.

As a Bayesian method, DPMUnc not only provides a clustering of the data, but can quantify the uncertainty about the clustering. This can be captured using the posterior similarity matrix (PSM) whose entry $\text{PSM}_{i,j}$ is the proportion of samples from the MCMC sampling in which observation *i* and observation *j* were placed in the same cluster [20]. We assessed how

well-calibrated the posterior similarity scores were, by binning PSM values into deciles, and examining the proportion of trait pairs in each bin that were truly in the same cluster. We found the two were generally similar, suggesting that DPMUnc was relatively well-calibrated (S3 Fig).

## Clustering immune-mediated diseases using GWAS summary statistics

Methodology has been developed to summarise genetic associations shared by multiple diseases using multi-dimensional polygenic scores, such as the summary of IMD by 13 such scores [21]. The paper also calculated these 13-dimensional scores for 1,230 independent GWAS datasets. We ran DPMUnc on 45 traits from these published data that showed significant differences to controls according to the original study (Fig 3). GWAS summary statistics include both an estimate of the strength of association and the uncertainty associated with that estimate. The uncertainty is propagated through to the scores, which is influenced heavily by sample size so each trait has similar uncertainty across all the scores, while uncertainty varies between traits (Fig 3).

Six clusters were identified, summarised as classic autoimmune diseases and most arthitides (A), a null cluster with observations lying close to zero or with sufficiently high uncertainty that they might truly lie near zero (B), classic autoinflammatory diseases and coeliac disease (C), multiple sclerosis (D), adult onset arthritis (E), and eosophilic granulomatisos granulomatosis with polyangiitis (EGPA) (F).

The posterior similarity matrix in S4 Fig shows relatively high confidence in the clusters. In particular, even with the uncertainty associated with the location of the rare disease EGPA, the two EGPA subtypes form their own cluster in a high percentage of the samples, perhaps reflecting their extreme location on PC13 which was shown to be related to eosinophils [21]. In contrast, whilst coeliac disease is clustered with other inflammatory bowel conditions (Crohn's disease and Ulcerative Colitis), it is much less confidently placed in this cluster than the other traits, perhaps reflecting that it is generally considered an autoimmune disease.

Multiple sclerosis (MS) is the only disease to cluster alone, which perhaps reflects that MS is unusual in having both autoinflammatory and autoimmune features [22], although the PSM does show both neuromyelitis optica (NMO) IgG$^-$ and systemic JIA have some non-zero probability of clustering with MS.

## Clustering patients by gene expression signatures

Gene expression signatures have been linked to several immune-mediated diseases, and we identified three such signatures from recent literature:

$Sig_{NK}$ contains genes enriched for expression in NK cells and has been associated with T1D [23]

$Sig_{IFN}$ contains genes in the type I interferon pathway, commonly found to show increased expression in SLE patients [24]

$Sig_{CD8}$ genes associated with CD8 T cell exhaustion, which have been shown to predict prognosis in inflammatory bowel disease [25, 26]

We attempted to cluster patients with a range of IMD in three gene expression datasets [24, 27, 28] (Table 1). For each individual, we summarised expression levels of each signature with a mean level and its associated uncertainty, as described in Methods. For some signatures, we anticipated association with specific diseases (such as the interferon signature and SLE) whilst for others we had less clear prior hypotheses. Whilst we did not expect every signature or every

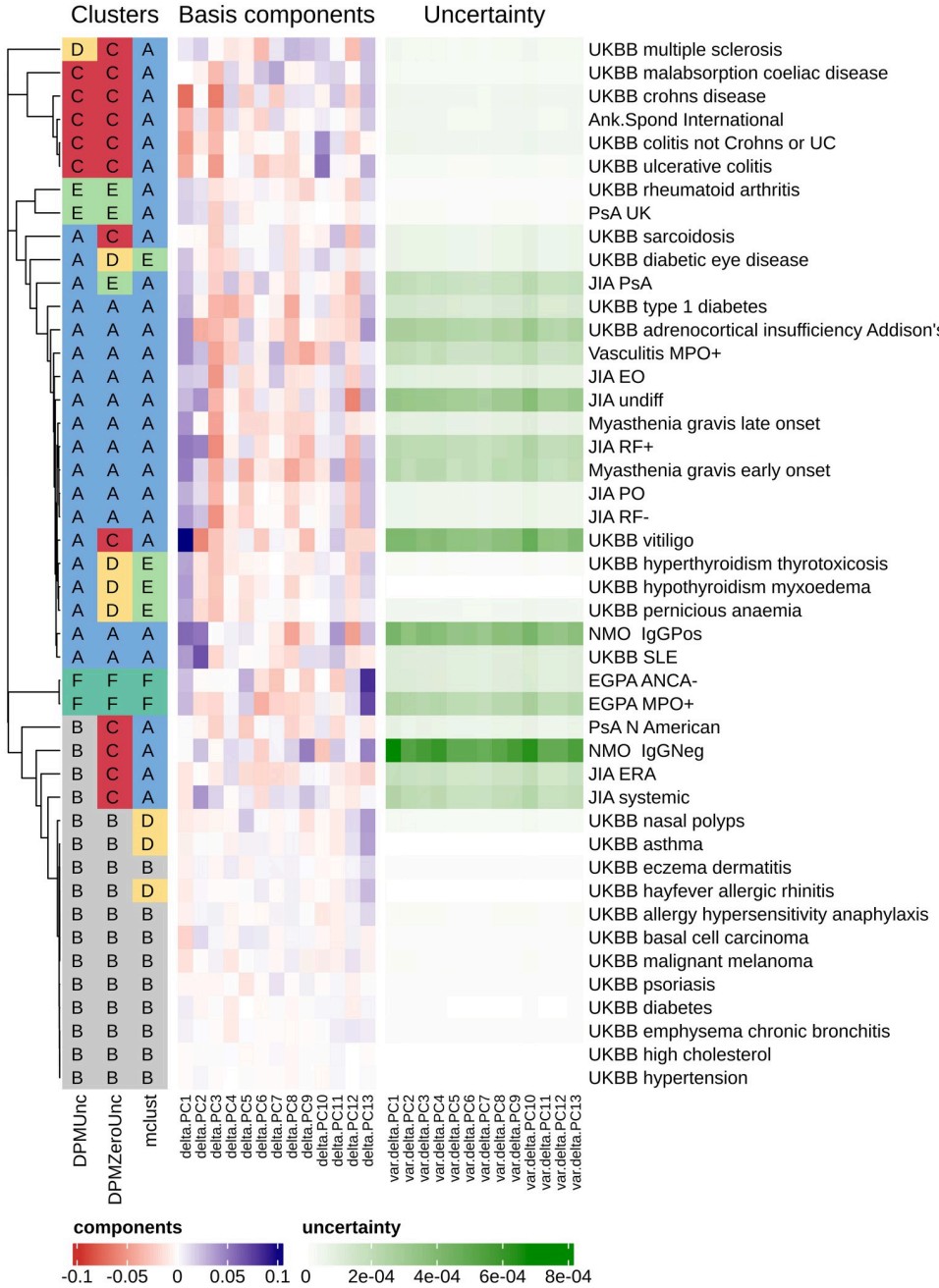

**Fig 3. Clusters inferred by DPMUnc, DPMZeroUnc and mclust alongside the polygenic GWAS scores and the associated uncertainty.** The clusters of DPMZeroUnc and mclust have been coloured to best match the colours of DPMUnc. The dendrogram on the left is formed by applying hierarchical clustering on the posterior similarity matrix from DPMUnc, using complete-linkage.

clustering to associate with disease, because any clusters found might reflect disease activity, sex or ethnicity, we considered that clusterings which discriminated between diseases would be meaningful, and so assessed the association of each clustering with disease using Fisher's exact test.

**Table 1. Sample counts in each gene expression dataset.** Counts for a subtype of a disease are in italics.

| | Ferreira [24] | Lyons [28] | Chaussabel [27] |
|---|---|---|---|
| Healthy controls (HC) | 88 | 25 | 12 |
| Systemic lupus erythematosus (SLE) | 25 | 13 | 103 |
| Juvenile idiopathic arthritis (JIA) | | | 47 |
| Melanoma (MEL) | | | 39 |
| Transplant (RET) | | | 37 |
| Type 1 diabetes (T1D) | 64 | | 20 |
| Vasculitis | | 30 | |
| *Microscopic Polyangiitis* (VWP) | | *8* | |
| *Wegener's Granulomatosis* (VMG) | | *22* | |
| Infection | | | 40 |
| *Infection (E. Coli)* (EColi) | | | *22* |
| *Infection (S. Aureus)* (SAureus) | | | *18* |
| Total | 177 | 68 | 298 |

We first used DPMUnc to cluster three datasets by each of the three individual signatures, i.e. nine clustering tasks overall, and found clusters in each dataset which showed association with disease in the majority of cases (7/9) (Fig 4). All datasets identified a "high" cluster for $Sig_{IFN}$ which contained a majority of SLE cases. Clustering by $Sig_{CD8}$ also found a "high" cluster which contained a majority of T1D cases in both datasets (Ferreira and Lyons). For $Sig_{NK}$, the Ferreira dataset identified a low cluster containing primarily T1D cases, as expected given the published associated of this signature with T1D [23], but whilst evidence for differential clustering by disease was found in Chaussabel, there was no enrichment of T1D in the "low" cluster. This difference may reflect time since diagnosis—T1D cases in Ferreira were predominantly newly diagnosed, while the samples in the original study that identified the signature were from children before and around the time of seroconversion. Only limited details are given about the Chaussabel T1D samples [27], but since by definition T1D cases are only newly diagnosed for a short period of time, it is likely these were longer standing cases. Thus this signature may represent an active disease process that resolves in long standing T1D cases.

Clustering using DPMUnc without uncertainty (DPMZeroUnc) produced very similar clusterings for 7/9 examples, but very different in two cases (S6 Fig). For $Sig_{CD8}$ in the Ferreira dataset, clustering without uncertainty produced two clusters instead of six, while for $Sig_{NK}$ in Ferreira, it found eight instead of two. Thus we saw no systematic difference in number of clusters by the two methods. However, while both examples produced clusterings that associated significantly with disease when including uncertainty ($p < 10^{-8}$), neither was associated with disease when uncertainty was ignored, suggesting that for these two examples accounting for uncertainty was required to reveal biologically meaningful clusterings.

We then clustered the Ferreira and Chaussabel datasets, which had shown association with each signature, across all three signatures simultaneously. We found that in the case of Ferreira, the multiple-signature clustering discriminated more powerfully between diseases than any individual signature clustering (Fig 5). However, in the case of Chaussabel, while the SLE group remained identifiable with high $Sig_{IFN}$ values, the T1D cluster that had been identified in $Sig_{CD8}$ clustering alone, was folded into the main cluster with the healthy controls.

When uncertainty was ignored (DPMZeroUnc), the Fereirra results were broadly similar (S7 Fig). Note, however, that we would not have considered multiple signature clustering on the basis of the DPMZeroUnc single dataset clusterings, as only one signature was significantly associated with disease. However, the Chaussabel dataset was clustered into only two groups,

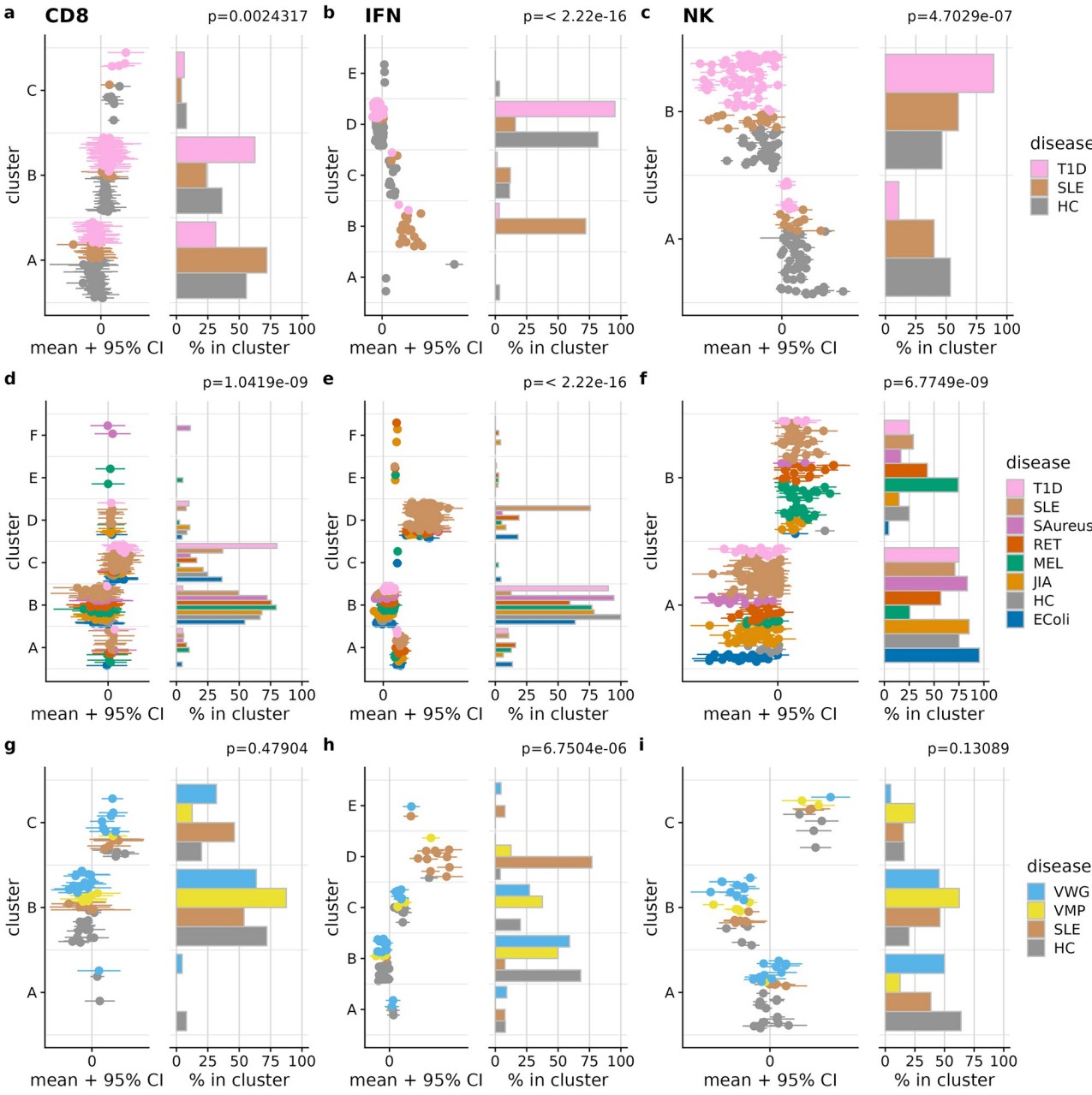

**Fig 4. Clustering of samples in 3 gene expression datasets (rows) according to 3 gene signatures (columns).** (a-c) Ferreira (d-f) Chaussabel (g-i) Lyons. In each panel, the left plot shows the observed data, and the right plot shows the fraction of individuals assigned to each cluster. The p value shown relates to the null hypothesis that cluster membership is independent of disease.

defined by IFN alone, with no distinction between samples with lower $Sig_{NK}$ values (groups F–I in Fig 5).

## Discussion

Taking uncertainty into account can alter clustering solutions. We illustrated with simulated data how the cluster mean inferred by DPMUnc is closer to the points that have lower uncertainty, whereas methods like mclust and k-means place the cluster mean close to the empirical

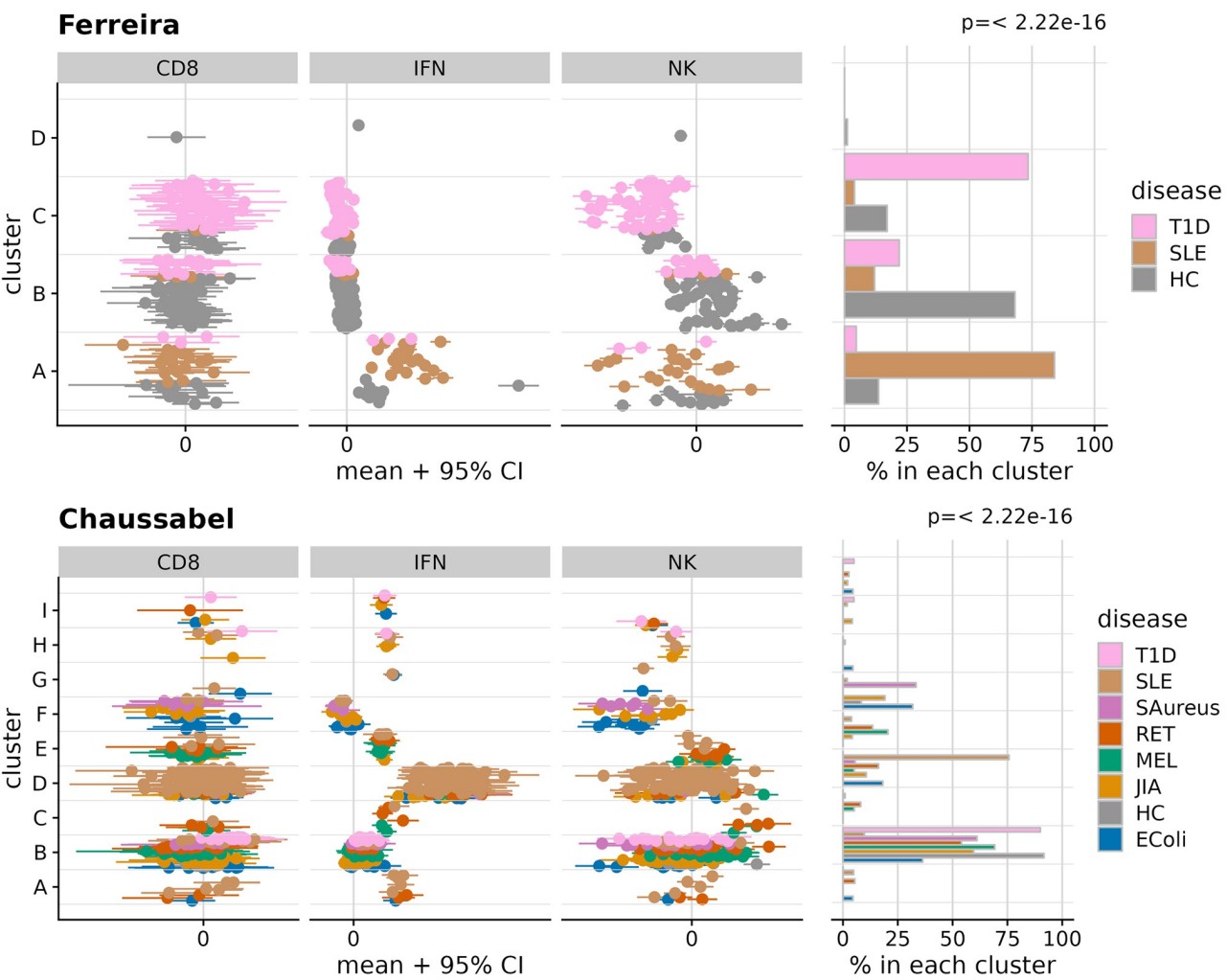

**Fig 5. Clustering of samples in 2 gene expression datasets (rows) using all 3 gene signatures.** In each panel, the left plot shows the observed data, and the right plot shows the fraction of individuals assigned to each cluster. The p value shown relates to the null hypothesis that cluster membership is independent of disease.

mean of all the points, and also how DPMUnc shifts the latent data points towards the inferred cluster mean, which results in a smaller cluster variance. In a more complicated dataset this could have knock-on consequences, perhaps excluding more distant points from joining the cluster, or splitting one cluster into two. On a range of simulated datasets, DPMUnc out-performed existing methods and the posterior similarity values were shown to be relatively well calibrated, so that if two points have posterior similarity of $p$, this roughly translates to probability $p$ that they are in the same cluster in the true clustering. We note that because DPMUnc assumes independence between features, its use is likely to be restricted to data that is relatively low dimensional. In the real data examples shown here, we assume there is limited dependence between the genetic features as these were derived using PCA and between the transcriptomic features because there is no overlap of genes between the different signatures.

Our clustering of diseases by the engineered GWAS features recapitulated known divisions and relationships in the immune-mediated diseases. It also suggests further uses for clustering with uncertainty, such as clustering individuals using sets of polygenic risk scores, where the uncertainty in the PRS coefficients could be exploited rather than ignored.

Gene signatures may be defined on one dataset with ideal experimental conditions, such as a timecourse study of the effect of interferon-$\beta$. To then use them on a new dataset with more complex structure, such as a dataset with patients with a mix of immune-mediated diseases, typically requires some form of dimension reduction such as WGCNA or PCA used on the new dataset, either using just those genes from the signature [24] or using all genes [26] with the hope that one WGCNA module or variable in PCA will coincide with the signature. Our proposed method of summarising a gene expression across a signature extends the possible uses of signatures, and since the signature may be weaker on a dataset with more complex structure, it may be crucial to take the variability of the signature into account, which DPMUnc allows. In two of the nine examples we examined, ignoring uncertainty led to clusterings which did not associate with disease status. In particular, it allows us to cluster patients simultaneously according to multiple signatures. In the case of the Ferreira dataset, this led to a clearer separation of the two disease and control groups, emphasizing the potential utility of considering multiple signatures, which provide multiple views of the same data.

## Methods

### Simulations

We simulated a variety of datasets with different characteristics to explore the behaviour of DPMUnc in comparison to other methods when the "truth" was known.

The simulated datasets each consist of thirty points spread across three clusters, and broadly follow the model used by DPMUnc outlined above. The simulation process includes two parameters which are varied between simulations: $U$ controls the magnitude of observation uncertainty and $N$ controls the level of noise around the cluster means, i.e. the levels of observational and biological variation respectively. The process for simulating data is:

1. Cluster means are the points $\mu_1 = (10, 0)$, $\mu_2 = (0, 10)$, $\mu_3 = (\alpha, \alpha)$ where $\alpha = 10\frac{1+\sqrt{3}}{2} \approx 14$ was chosen so that the cluster means are equidistant.

2. Points were independently and uniformly assigned to clusters i.e. $c_i \overset{\text{i.i.d.}}{\sim} \text{Uniform}\{1, 2, 3\}$ for $i = 1, \ldots, 30$.

3. Latent observations were simulated around the cluster mean, with variance $N$ i.e. $z_i \mid c_i = k \sim \mathcal{N}(\mu_k, N)$

4. Uncertainty for the points was simulated using $\sigma_i^2 \overset{\text{i.i.d.}}{\sim} \Gamma(k = 1/2, \theta = 2U)$.

5. Observed data points were simulated about the latent points: $x_i \sim \mathcal{N}(z_i, \sigma_i^2)$

We simulated for 100 different seeds each combination of $U = 5, 7, 10, 13$ and $N = 4, 9, 12, 16$. A grid of simulated datasets is shown in S5 Fig showing the range of the datasets.

We ran DPMUnc and DPMZeroUnc with 10 thousand iterations, saving only every 10th iteration to avoid autocorrelation and discarding the first half of the iterations as burn-in.

### Prior for cluster parameters

We used the prior hyperparameters $\alpha_0 = 2$, $\kappa_0 = 0.5$. $\mu_0$ is set to be the empirical mean of all the data. $\beta_0$ is set to be 0.2 multiplied by the variance of the variable, thus scaling $\beta_0$ by the variance of the variable. In datasets with multiple variables, we use the mean of the variances to scale $\beta_0$. If a dataset had variables on vastly different scales, this would not be an appropriate choice of prior but all the datasets in this paper have variables on comparable scales.

## Comparing to other methods

We used simulated data to examine the performance of DPMUnc. Simulated data provides access to the true latent observations and so we could use the clustering ability on the latent data as a guide to the best performance possible by DPMUnc: no matter how good the adjustment made for uncertainty of the points is, DPMUnc cannot be expected to do better than its performance on the true latent data. We also used two widely used clustering methods as benchmarks, using the adjusted rand index (ARI) [29] to quantify the accuracy.

k-means [6, 7] can be shown to be equivalent to a GMM with spherical clusters. The R package *cluster* [30] provides a function *clusGap* which calculates a goodness of clustering measure called the gap statistic [18], which measures within-cluster similarity. We use this measure to choose the optimal value of $k$, the number of clusters. We ran kmeans with 25 random starts to avoid convergence to a local optimum.

mclust [8] is a Gaussian Mixture Model (GMM) which does not account for uncertainty and which requires $K$ to be specified. We ran mclust with default settings and chose $K$ by optimising the Bayesian Information Criterion as recommended by the authors.

The Bhattacharyya distance [19] is a measure of similarity between two probability distributions. We take each observation as the centre and the variance-covariance matrix as the variance of a multivariate normal. We then cluster the observations using hierarchical clustering with the R function hclust, choosing K as for k-means.

We ran fuzcluster [13] using the fuzcluster function in the R package sirt. We estimate $K$ as for kmeans.

We implemented the strategy of representative clustering of uncertain data [12] in R, and used it to extend kmeans and mclust for uncertain data.

We ran DPMUnc with the simulated data and with adjusted versions of the datasets where the uncertainty estimates were shrunk to 0. Thus the latent variables are essentially fixed to be equal to the observed data points throughout. We call this version of the method DPMZeroUnc.

## GWAS summary statistics features

We downloaded Table S10 from [21] and kept the 186 traits which differed significantly from a control dataset (FDR<0.01), and focused on disease traits. To avoid near duplicates, we retained only one set of results from UK Biobank (UKBB) by excluding the GeneAtlas analysis, and retaining the Neale analyses which had a greater representation of rarer immune traits, leaving us with 45 traits in our final dataset.

We ran DPMUnc and DPMZeroUnc with 100 million iterations, saving only every 1000th iteration to avoid autocorrelation and discarding the first half of the iterations as burn-in.

**Preprocessing gene expression data.** We first used a variance stabilising transformation [31]*vsn2* to minimise the relationship between average gene expression and the variance of gene expression (S8 Fig). We then used median absolute deviation (MAD) scale normalization to make the distribution of expression values similar between genes, dividing all values for gene *g* by the MAD of the control samples for gene *g*.

## Gene expression signatures

The genes representing each signature were identified as follows

*Sig*$_{NK}$ consists of 87 genes (Supplementary File S6 from [23]).

$Sig_{IFN}$ consists of 56 genes identified through differential expression analysis using interferon stimulation of PBMCs and marked as discriminatory in SLE according to Supplementary Table 2 [24].

$Sig_{CD8}$ consists of 12 genes from module 8 and module 9 in [25] Supplemental Table 1 which showed reduced expression with log fold change $<-3$ only in exhausted CD8 T cells. The signature was defined in mice and we used BioMart to convert from murine (GRCm39) genes to human (GRCh38.p13) genes [32, 33].

The selected signature genes are listed in full in S1 Table and were summarised for each sample by two values, the mean and the variance of the normalised expression value, with the summary measures taken across all genes in a signature.

We ran DPMUnc and DPMZeroUnc with 20 million iterations, saving only every 500th iteration to avoid autocorrelation and discarding the first half of the iterations as burn-in.

## MCMC summarisation approach

Whilst DPMUnc and DPMZeroUnc provide samples from a posterior distribution, it is useful to find a single summary clustering for downstream analyses. We first calculate the posterior similarity matrix (PSM). Each entry $PSM_{i,j}$ is the proportion of posterior samples in which observation $i$ and observation $j$ were placed in the same cluster. Secondly, we calculate the clustering of the PSM that optimises the posterior expected adjusted rand index (PEAR) with the true clustering (maxpear [20]), i.e. the clustering that seems the most compatible with the true clustering. This requires a search strategy to explore the space of possible clusters, and we use all possible clusters found via complete hierarchical clustering, using $1 - PSM$ as a distance matrix, using the mcclust and mcclust.ext packages in R.

## Supporting information

**S1 Text. Supplementary text describing the mathematical details of the model, illustrative examples and runtimes.**
(PDF)

**S1 Fig. MCMC algorithm used by DPMUnc.**
(TIFF)

**S2 Fig. Estimates of $K$ for different methods across simulated datasets. In all cases true K is 3.**
(TIFF)

**S3 Fig. Calibration of posterior similarity scores.** The green line shows perfect calibration, where the posterior similarity score is the same as the proportion of pairs that are in the same cluster in the true clustering. For each collection of pairs of points with posterior similarity score within some bin, the proportion of pairs which are in the same cluster in the true clustering is shown by a dot.
(TIFF)

**S4 Fig. Posterior similarity matrix for DPMUnc's clustering.** Both rows and columns correspond to traits and each entry in the grid shows the proportion of samples from the MCMC chain in which the two traits were placed in the same cluster.
(TIFF)

**S5 Fig. Example simulated datasets with varying levels of true noise $N$ and observation uncertainty $U$.** The latent data are empty circles, with arrows pointing to the observed values,

which are represented by filled circles. The points are coloured by cluster membership. The left column has lowest observation uncertainty, so that the observations are close to their latent points, whereas the right column has high observation uncertainty. The top row has lowest cluster noise, with most latent data lying close to the cluster mean whereas the bottom row has higher cluster noise, with latent data more spread out.
(TIFF)

**S6 Fig. Clustering of samples in 3 gene expression datasets (rows) according to 3 gene signatures (columns) using DPMZeroUnc.** (a-c) Ferreira (d-f) Chaussabel (g-i) Lyons. In each panel, the left plot shows the observed data, and the right plot shows the fraction of individuals assigned to each cluster. The p value shown relates to the null hypothesis that cluster membership is independent of disease. This can be compared to Fig 4 where clustering used DPMUnc to allow for uncertainty.
(TIFF)

**S7 Fig. Clustering of samples in 2 gene expression datasets (rows) using all 3 gene signatures using DPMZeroUnc.** In each panel, the left plot shows the observed data, and the right plot shows the fraction of individuals assigned to each cluster. The p value shown relates to the null hypothesis that cluster membership is independent of disease. This can be compared to Fig 5 where clustering used DPMUnc to allow for uncertainty.
(TIFF)

**S8 Fig. Relationship between mean and standard deviation in the Ferreira dataset, before (a) and after (b) vsn2 transformation.**
(TIFF)

**S1 Table. Genes in each signature used in the gene expression datasets.**
(CSV)

# Acknowledgments

This work was performed using resources provided by the Cambridge Service for Data Driven Discovery (CSD3) operated by the University of Cambridge Research Computing Service (www.csd3.cam.ac.uk), provided by Dell EMC and Intel using Tier-2 funding from the Engineering and Physical Sciences Research Council (capital grant EP/P020259/1), and DiRAC funding from the Science and Technology Facilities Council (www.dirac.ac.uk).

# Author Contributions

**Data curation:** Kath Nicholls.

**Investigation:** Kath Nicholls, Chris Wallace.

**Methodology:** Paul D. W. Kirk.

**Supervision:** Chris Wallace.

**Visualization:** Kath Nicholls, Chris Wallace.

**Writing – original draft:** Kath Nicholls, Chris Wallace.

**Writing – review & editing:** Paul D. W. Kirk.

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
