## [Decision Letter · Decision Letter 0]

23 Apr 2023

Dear Ms Nicholls,

Thank you very much for submitting your manuscript "Bayesian clustering with uncertain data" for consideration at PLOS Computational Biology.

As with all papers reviewed by the journal, your manuscript was reviewed by members of the editorial board and by several independent reviewers. In light of the reviews (below this email), we would like to invite the resubmission of a significantly-revised version that takes into account the reviewers' comments. Specifically, we would like you to address the major concerns about comprehensive comparisons of the method to relevant baselines and other methods, scalability of the method for larger instances and justify the novelty claims related to grouping genes with respect to signatures. 

We cannot make any decision about publication until we have seen the revised manuscript and your response to the reviewers' comments. Your revised manuscript is also likely to be sent to reviewers for further evaluation.

Sincerely,

Ferhat Ay, Ph.D

Academic Editor

PLOS Computational Biology

Jian Ma

Section Editor

PLOS Computational Biology

Reviewer's Responses to Questions

**Comments to the Authors:**

Reviewer #1: Authors extended the use of Dirichlet Process Gaussian Mixture models to incorporate the uncertainty associated with each sample. And they proposed Dirichlet Process Matrix Mixtures with Uncertainty (DPMUnc) as a nonparametric clustering framework. Authors conducted a simulation study that compared DPMUnc with existing methods, and further demonstrated that DPMUnc was effective in classifying immune-mediated diseases using GWAS summary statistics and clustering patients based on relevant gene signatures. The manuscript is well written and easy to follow, with detailed derivations included in the supplementary information. All the analysis pipelines are well-documented, ensuring the reproducibility of the study.

The use of uncertainty as a means of improving cluster assignment is an interesting problem, and the authors’ proposed model appears to be theoretically sound. However, I have some concerns about the comparisons. Authors compared DPMUnc with Gaussian Mixture Model (GMM implementation) and k-means which are two baseline methods, while these two methods are not explicitly designed to handle data uncertainty, which may make the comparisons less persuasive. Moreover, certain details about the experiments need to be clarified.

Major concerns

1. The introduction section may benefit from a brief discussion on clustering the uncertain data, and authors could consider expanding the section to provide a more comprehensive overview. Additionally, there are alternative approaches clustering data with uncertainty that authors could include in the comparison. For instance, a few algorithms can be found via the following link: https://github.com/fgullo/jcludata.

2. In the simulation study, I am wondering if mclust and kmeans were initialized from multiple starting points to avoid local optimal solutions.

3. Authors clustered immune-mediated diseases with the GWAS statistics (Figure 3). DPMUnc and DPMZeroUnc disagree on the the diseases where the variance is relatively small, e.g., UKBB hyperthyroidism thyrotoxicosis, UKBB hypothyroidism myxoedema and UKBB pernicious anaemia. Could you elaborate on this observation?

4. Authors run DPMUnc to classify public gene expression dataset using relevant signatures. It would be helpful to include DPMZeroUnc and mclust to compare with DPMUnc and report corresponding ARI values. These could be included as part of the supplementary information. Another natural baseline here is to use mclust/kmeans to cluster all the genes included in the signature set without aggregation. This is a common practice to address similar clustering problems, and it would be compelling to show that DPMUnc can outperform this standard clustering pipeline.

Minor concerns

1. Authors may consider broadening the scope of possible applications of the DPMUnc method as uncertainty commonly exists in most of the biological measurements. For example

(1) Authors could investigate if DPMUnc can improve upon the problem of clustering data based on multiple uncertain experts.

(2) Authors could experiment with network clustering as every edge/interaction was measured with a confidence score indicating uncertainty [1]

(3) Authors could explore the cell type annotation problem when dealing single-cell multiomic data. In this case, one cell could be assigned different cell types based on different modalities, e.g., RNA-seq and ATAC-seq. DPMUnc may be useful for integrating the annotation and clustering the single-cell data more accurately [2].

2. The uncertainty variable (variance) only influences the latent variable sampling step. Therefore, DPMZeroUnc should be equivalent as the standard Dirichlet Process Gaussian Mixture. It may be beneficial for authors to compare DPMZeroUnc with existing implementations of Dirichlet Process Gaussian Mixture models, such as sklearn, to verify if the convergence behavior is similar.

3. Sampling algorithms in general are computationally intensive. Authors can include some basic statistics on the running time and include a short discussion in the last section.

4. Supplementary figures S1 and S2 are really intuitive and helpful. It would enhance the clarity of the figures and make the respective legends more reader friendly if different panels can be labeled by letters.

Reference

1. Jonsson, Pall F., and Paul A. Bates. "Global topological features of cancer proteins in the human interactome." Bioinformatics 22.18 (2006): 2291-2297.

2. Azimuth, https://azimuth.hubmapconsortium.org/

Reviewer #2: The review is uploaded as an attachment. See "Comments to authors.pdf".

Reviewer #3: Review is attached as a pdf

**Have the authors made all data and (if applicable) computational code underlying the findings in their manuscript fully available?**

Reviewer #1: Yes

Reviewer #2: Yes

Reviewer #3: Yes

PLOS authors have the option to publish the peer review history of their article (what does this mean?). If published, this will include your full peer review and any attached files.

Reviewer #1: No

Reviewer #2: No

Reviewer #3: No
---

## [Decision Letter · Decision Letter 1]

14 Oct 2023

Dear Ms Nicholls,

Thank you very much for submitting your manuscript "Bayesian clustering with uncertain data" for consideration at PLOS Computational Biology.

As with all papers reviewed by the journal, your manuscript was reviewed by members of the editorial board and by several independent reviewers. In light of the reviews (below this email), we would like to invite the resubmission of a version that takes into account the reviewers' comments. As you will see, there are only a few major concerns remaining but minor concerns such as leaving out TODO mark etc. also need to be addressed carefully.

We cannot make any decision about publication until we have seen the revised manuscript and your response to the reviewers' comments. Your revised manuscript is also likely to be sent to reviewers for further evaluation.

Sincerely,

Ferhat Ay, Ph.D

Academic Editor

PLOS Computational Biology

Jian Ma

Section Editor

PLOS Computational Biology

Reviewer's Responses to Questions

**Comments to the Authors:**

Reviewer #1: I appreciate the authors' comprehensive response, which effectively addressed most of my concerns. Overall, my impression is positive. While I acknowledge the arguments presented in response to my primary concern #4, I would like to underscore the importance of comparing the diverse outcomes yielded by different methods on the real datasets. Simulation experiments offer a controlled and constrained environment for showcasing the superior performance of the proposed method. The objective of real experiments, however, is not to identify the "best" model, given the potential variability in their outputs. Rather, it is to address the question: what novel insights can be exclusively derived through the application of DPMUnc as opposed to alternative approaches? This inquiry aims to provide an intuitive grasp of DPMUnc's behavior within a practical context. I hope this clarification will alleviate any confusion stemming from my initial inquiry.

Upon careful review of the response, I observed several placeholders labeled as TODO within the response document. I strongly recommend a thorough examination prior to finalizing and submitting the document.

Reviewer #2: The review is uploaded as an attachment.

**Have the authors made all data and (if applicable) computational code underlying the findings in their manuscript fully available?**

Reviewer #1: None

Reviewer #2: None

PLOS authors have the option to publish the peer review history of their article (what does this mean?). If published, this will include your full peer review and any attached files.

Reviewer #1: No

Reviewer #2: No
---

## [Decision Letter · Decision Letter 2]

8 Jul 2024

Dear Dr. Wallace,

We are pleased to inform you that your manuscript 'Bayesian clustering with uncertain data' has been provisionally accepted for publication in PLOS Computational Biology.

Best regards,

Ferhat Ay, Ph.D

Academic Editor

PLOS Computational Biology

Jian Ma

Section Editor

PLOS Computational Biology

Reviewer's Responses to Questions

**Comments to the Authors:**

Reviewer #1: My concerns have been properly addressed, and I have no further comments on the methods or the results part. A few minor edits need attention later:

1. There are placeholders in the latest response as “Fig SX and Fig SX”, which should refer to S6 Fig and S7 Fig.

2. The proposed updates to the Discussion, specifically “Our proposed method of summarizing a gene expression across a signature extends the possible uses of signatures, and since the signature may be weaker on a dataset with more complex structure, it is crucial to take the variability of the signature into account, which DPMUnc allows.” are confusing. Authors may revise this sentence for clarity.

3. Currently, S6 Fig lacks dataset names as panel titles or corresponding descriptions in the legend (similar to Fig. 4) to describe the correspondence between panels and the datasets included.

**Have the authors made all data and (if applicable) computational code underlying the findings in their manuscript fully available?**

Reviewer #1: None

PLOS authors have the option to publish the peer review history of their article (what does this mean?). If published, this will include your full peer review and any attached files.

Reviewer #1: No

---

## [Editor Report · Acceptance letter]

26 Aug 2024

PCOMPBIOL-D-22-01876R2 

Bayesian clustering with uncertain data

Dear Dr Wallace,

I am pleased to inform you that your manuscript has been formally accepted for publication in PLOS Computational Biology. Your manuscript is now with our production department and you will be notified of the publication date in due course.

With kind regards,

Olena Szabo
